

# Why do we transition from walking to running? Energy cost and lower leg muscle activity before and after gait transition under body weight support

Daijiro Abe[1], Yoshiyuki Fukuoka[2] and Masahiro Horiuchi[3]

[1] Center for Health and Sports Science, Kyushu Sangyo University, Fukuoka, Japan
[2] Faculty of Health and Sports Science, Doshisha University, Kyotanabe, Kyoto, Japan
[3] Division of Human Environmental Science, Mt. Fuji Research Institute, Fujiyoshida, Yamanashi, Japan

## ABSTRACT

**Background:** Minimization of the energetic cost of transport (CoT) has been suggested for the walk-run transition in human locomotion. More recent literature argues that lower leg muscle activities are the potential triggers of the walk-run transition. We examined both metabolic and muscular aspects for explaining walk-run transition under body weight support (BWS; supported 30% of body weight) and normal walking (NW), because the BWS can reduce both leg muscle activity and metabolic rate.

**Methods:** Thirteen healthy young males participated in this study. The energetically optimal transition speed (EOTS) was determined as the intersection between linear CoT and speed relationship in running and quadratic CoT-speed relationship in walking under BWS and NW conditions. Preferred transition speed (PTS) was determined during constant acceleration protocol (velocity ramp protocol at $0.00463 \text{ m·s}^{-2} = 1 \text{ km·h}^{-1}$ per min) starting from $1.11 \text{ m·s}^{-1}$. Muscle activities and mean power frequency (MPF) were measured using electromyography of the primary ankle dorsiflexor (*tibialis anterior*; TA) and synergetic plantar flexors (calf muscles including *soleus*) before and after the walk-run transition.

**Results:** The EOTS was significantly faster than the PTS under both conditions, and both were faster under BWS than in NW. In both conditions, MPF decreased after the walk-run transition in the dorsiflexor and the combined plantar flexor activities, especially the *soleus*.

**Discussion:** The walk-run transition is not triggered solely by the minimization of whole-body energy expenditure. Walk-run transition is associated with reduced TA and *soleus* activities with evidence of greater slow twitch fiber recruitment, perhaps to avoid early onset of localized muscle fatigue.

Corresponding author
Daijiro Abe, abed@ip.kyusan-u.ac.jp

## INTRODUCTION

Human terrestrial locomotion involves two major gait patterns: walking and running. A biological benefit of these two patterns is in facilitation of economical energy

expenditure (EE) (*Margaria, 1938*). When humans gradually increase their walking speed, they reach a speed at which they naturally change their gait pattern from walking to running, and it has been suggested that such a walk-run gait transition minimizes their metabolic rate (*Cavagna & Kaneko, 1977*; *Minetti, Ardigò & Saibene, 1994*). In walking, there is a *U*-shaped relationship between the energetic cost of transport per unit distance (CoT) and speed (*s*); for running, the relationship is approximately constant (*Abe, Fukuoka & Horiuchi, 2015*). Consequently, there is an intersection between the CoT-*s* relationship for walking and running; this is known as the energetically optimal transition speed (EOTS). However, the EOTS calculated in this way has been shown to be approximately 5–6% faster than the "preferred" transition speed (PTS) at which humans make the walk-run transition (*Hreljac, 1993*; *Rotstein et al., 2005*; *Tseh et al., 2002*), so factors other than a minimization of the metabolic rate are needed to explain the trigger for the walk-run transition in human locomotion.

An abrupt increase in muscle activity of the dorsiflexor (*tibialis anterior*; TA) has been observed when participants walked at a speed close to the PTS (*Hreljac, 1995*; *Hreljac et al., 2008*; *Prilutsky & Gregor, 2001*; *Bartlett & Kram, 2008*; *Malcolm et al., 2009*; *Shih et al., 2016*), and our recent studies showed muscle activity of the TA decreased when participants switched from walking to running at the EOTS (*Abe, Fukuoka & Horiuchi, 2017*; *Abe et al., 2018*). In association with a decrease in TA activity when switching the gait pattern, mean power frequency (MPF; Hz) of the TA became lower, suggesting that more Type I muscle fibers were recruited in the TA during running than walking at the EOTS. These findings suggest that muscle activity of the TA could be a trigger of the walk-run transition. Conversely, other reports have proposed that the medial head of *gastrocnemius* muscle (MG) of the plantar flexors, an antagonist of the TA, plays a key role in the walk-run transition (*Farris & Sawicki, 2012*; *Neptune & Sasaki, 2005*), because ankle force production by the MG was observed to decrease in proportion to the increase in walking speed. Thus, the trigger for the walk-run transition remains under debate.

One approach to resolve this matter is to perform experiments using simulated reduced gravity and/or body weight support (BWS) conditions. This reduces not only leg muscular activity but also the required EE during walking (*Grabowski, 2010*; *McGowan, Neptune & Kram, 2008*, *2009*; *Pavei, Biancardi & Minetti, 2015*) and running (*Raffalt, Hovgaard-Hansen & Jensen, 2013*; *Teunissen, Grabowski & Kram, 2007*). Because EE is potentially reduced by the decreased plantar flexor activity (*Hamner, Seth & Delp, 2010*; *Neptune, Zajac & Kautz, 2004*), it would be expected that the muscular activity of the plantar flexors would be lower in BWS than in normal walking (NW) conditions. In that case, the PTS and/or EOTS would be faster in BWS than in NW conditions. Summarizing the results of the above previous studies, we hypothesized that the PTS and/or EOTS would be faster in BWS compared to NW. We also hypothesized that TA activities would decrease after the walk-run transition under both conditions and that muscular activity of the plantar flexors would decrease in BWS than in NW, while it remained constant regardless of walk-run transition. The purpose of this study was to investigate the muscular activity and the motor unit recruitment patterns of the dorsiflexor and plantar flexors before and after the walk-run transition under BWS and NW conditions.

| Table 1 Physical characteristics of each participant. | | | | | |
|---|---|---|---|---|---|
| Participant | Age (years) | Stature (m) | BW (kg) | BF (%) | BMI (kg·m$^{-2}$) |
| A | 19.6 | 1.696 | 72.0 | 19.8 | 25.0 |
| B | 21.1 | 1.700 | 66.4 | 19.8 | 23.0 |
| C | 21.7 | 1.638 | 56.5 | 16.2 | 21.1 |
| D | 19.8 | 1.703 | 54.2 | 10.6 | 18.7 |
| E | 19.8 | 1.635 | 62.6 | 20.2 | 23.4 |
| F | 20.0 | 1.650 | 70.0 | 25.1 | 25.7 |
| G | 21.2 | 1.665 | 66.5 | 16.5 | 24.0 |
| H | 19.5 | 1.684 | 56.9 | 13.0 | 20.1 |
| I | 19.9 | 1.740 | 81.0 | 24.3 | 26.8 |
| J | 19.3 | 1.634 | 56.7 | 17.6 | 21.2 |
| K | 20.6 | 1.736 | 49.2 | 10.4 | 16.3 |
| L | 19.9 | 1.746 | 55.8 | 13.6 | 18.3 |
| M | 21.3 | 1.620 | 56.4 | 17.6 | 21.5 |
| Average | 20.3 | 1.681 | 61.9 | 17.3 | 21.9 |
| SD | 0.8 | 0.044 | 8.9 | 4.6 | 3.1 |

**Note:**
BW, body weight; BF, percent of body fat; BMI, body mass index, respectively.

## MATERIALS AND METHODS

### Participants

Thirteen young healthy males participated in the present study. Their physical characteristics are summarized in Table 1. In accordance with the Declaration of Helsinki, all participants were provided all information about the purposes, benefits, possible risks, and experimental protocols. A written informed consent was obtained from all participants. An ethical committee established in Kyushu Sangyo University approved all procedures of this study (H28-0001-1).

### Body weight support

BWS was established with a custom-made body suspension apparatus that lifted the participant's torso via an elastic rehabilitation harness (Fig. 1). This was installed around a motor-driven treadmill (Well Load 200E; Takei Scientific Instruments Co. Ltd., Niigata, Japan). For the BWS condition, we chose a 30% decrement in the participant's body weight (BW), equivalent to that used in some previous studies (Grabowski, 2010; McGowan, Neptune & Kram, 2008, 2009; Raffalt, Hovgaard-Hansen & Jensen, 2013; Teunissen, Grabowski & Kram, 2007). A force transducer (TSA-110; Takei Scientific Instruments Co. Ltd., Niigata, Japan) was located between a control box and spring (with 30 cm free length and a spring constant of 5.7 kg·cm$^{-1}$) to slide a rigid main frame (see Video S1). Before the participant walked or ran, he stood stationary on the treadmill while wearing the harness, and 30% of his BW was lifted by the spring system. The set-up allowed normal leg swing.
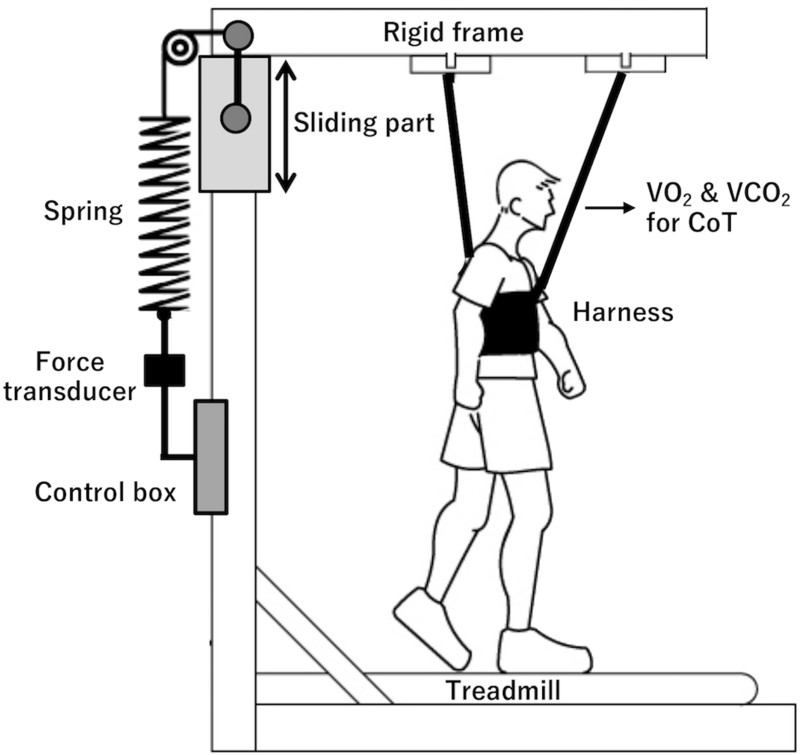

**Figure 1 Body weight support apparatus.** A schematic illustration of the participants walking with a custom-made body weight support (BWS) apparatus. This setting ensured that the BWS apparatus applied purely vertical force without disturbing leg swing motion.

## Protocols and determination of EOTS

All participants continuously walked on a motor-driven treadmill at four walking speeds (1.33, 1.56, 1.78, and 2.00 m·s$^{-1}$) on a level (±0%) gradient. After measuring the metabolic cost of walking, the participants took a sitting rest for 7–8 min. They started at three running speeds (2.00, 2.22, and 2.44 m·s$^{-1}$) with 1 min standing rests between the running stages. Each walking and/or running speed was kept constant for 4 min. At all stages, the participants chose their preferred step frequencies. They wore underwear, shirts, socks, and the same model of shoes in the appropriate sizes (Wave Wing, Mizuno, Japan).

Oxygen uptake (VO$_2$; mL·kg$^{-1}$·s$^{-1}$) and carbon dioxide output (VCO$_2$; mL·kg$^{-1}$·s$^{-1}$) were continuously measured with a computerized breath-by-breath system (AE-310S; Minato Ltd., Tokyo, Japan). Calibrated gas concentrations (O$_2$ 15.22%, CO$_2$ 5.17%, and N$_2$ 79.61%) and room air were used for the calibration of the gas analyzer. An average VO$_2$ and VCO$_2$ for the final 2 min at each speed was used to calculate the CoT values using a following equation (*Brouwer, 1957*).

$$\text{CoT (J·kg}^{-1}\text{·m}^{-1}) = \frac{4.186 \times (3.869 \times \text{VO}_2 + 1.195 \times \text{VCO}_2)}{s} \tag{1}$$

where the *s* is the walking or running speed. The CoT values were compared at each gait speed between both conditions. The relationship between the CoT values and

walking speeds was approximated with a quadratic equation (*Abe, Fukuoka & Horiuchi, 2015*):

$$\text{CoT}(s) = as^2 + bs + c \tag{2}$$

where the coefficients a, b, and c are determined by the least squares regression with data obtained from four walking speeds. The relationship between CoT values and running speeds was approximated using a linear equation (*Abe, Fukuoka & Horiuchi, 2015*). The running CoT values was described as follows:

$$\text{CoT}(s) = ps + q \tag{3}$$

where the coefficients p and q are determined by the least squares regression with the CoT values from three running speeds. An extrapolation of the quadratic equation Eq. (2) meets a linear regression line Eq. (3), and then an intersection (EOTS) of these equations was obtained. This extrapolation has been normally used for determining the EOTS in other previous studies (*Abe, Fukuoka & Horiuchi, 2015*; *Hreljac, 1993*; *Minetti, Ardigò & Saibene, 1994*; *Rotstein et al., 2005*; *Tseh et al., 2002*). Thus, the EOTS is an extrapolated value when the Eqs. (2) and (3) are equal. Rearranging Eqs. (2) and (3):

$$as^2 + (b - p)s + (c - q) = 0 \tag{4}$$

A following formula gives two solutions of Eq. (4), thus, only the faster one is regarded as the EOTS (*Abe, Fukuoka & Horiuchi, 2015*).

$$\text{EOTS}\,(\text{m·s}^{-1}) = \frac{-(b - p) \pm \sqrt{(b - p)^2 - 4a(c - q)}}{2a} \tag{5}$$

## EMG measurements at the PTS

The participant sat down on a chair to allow placement of the pre-amplified active surface EMG electrodes (BA-U410m; Nihon Santeku Co. Ltd., Osaka, Japan) on *tibialis anterior* (TA), medial and lateral heads of MG and LG, and *soleus* (SOL). Before electrode placement, the skin was shaved and wiped with alcohol for an exfoliation. The electrodes were secured using surgical tape to avoid disturbing the locomotor tasks. A foot sensor (PS-20KASF4; Kyowa Electronic Instruments Co. Ltd., Tokyo, Japan) was inserted into a right shoe to count number of steps, and its signal was amplified with a signal conditioner (CDV-700A; Kyowa Electronic Instruments Co. Ltd., Tokyo, Japan). This foot sensor also allowed us to visually detect what the participant transitioned from walking to running, because the vertical force suddenly increased when the participant started running. The vertical force itself should be ignored, because the foot sensor is cushioned by the shoe sole and plantaris muscle. The participant was asked to start walking at 1.11 m·s$^{-1}$ for 1 min at a freely chosen step frequency. After this steady-state walking, the treadmill speed was gradually increased at an acceleration of 0.00463 m·s$^{-2}$ (1 km·h$^{-1}$ per min) in a modified incremental ramp manner. The speed at which the walk-run transition occurred was considered to be the PTS. The participant was asked to keep running for 1 min after the walk-run transition. EMG was recorded throughout the whole ramp protocol. The BWS condition was tested before the NW condition because it took several minutes to put on the harness without disturbing the wired EMG

electrodes and bio-amplifier attached to the participant's waist. Data for one participant was excluded from the analysis because of an artifact in the EMG. The analyzed time duration was set as 12 s before and after the walk-run transition; in practice, it was $11.972 \pm 1.562$ s, equivalent to $13.1 \pm 1.9$ steps when walking or $15.2 \pm 2.3$ steps when running.

The EMG signals were amplified with a gain of ×1000 (BA 1104B; Digitex Lab Co. Ltd., Tokyo, Japan). Sampling frequency was set at 2 kHz, and a band-pass filter (8–500 Hz) was applied for the EMG signals. All signals from each sensor were simultaneously recorded with software (MaP 1038 ver.7.4; Nihon Santeku Co. Ltd., Osaka, Japan). Subsequently, a fast Fourier transform was applied to the stored EMG signals to obtain MPF; Hz. MPF reflects motor unit recruitment pattern in the exercising muscles (*De Luca, 1997*; *Wakeling, 2004*). An alteration of the MPF values between walking and running allows us to evaluate muscle fiber recruitment pattern in each gait. The sum of the rectified EMG (µV·s) was normalized by measured time (s) and number of steps to cancel the effects of step frequency (*Abe, Fukuoka & Horiuchi, 2017*; *Abe et al., 2018*). The EMG values obtained were further normalized to those obtained at 1.11 m·s$^{-1}$ under each condition.

## Statistical analysis

Data were presented as mean ± SD. The CoT values were compared with two-way repeated measures analysis of variance (ANOVA) within participants (BW × speed) using online software (ANOVA 4 © 2002 Kiriki Kenshi; https://www.hju.ac.jp/~kiriki/anova4/). Two-way repeated measures of ANOVA was also applied for comparisons among the EOTS and PTS obtained under both gravities. Each muscle activity and MPF were also compared in the same way. The present study was exploratory in design, and the minimum detectable $F$ value was 4.747 when using G*Power 3.1 (*Faul et al., 2007*) with an actual number of 13 participants. A partial eta-squared value ($\eta^2$) was also presented (*Cohen, 1988*). If a significant $F$ value was obtained on the dependent variables, Ryan's post hoc test was applied to the appropriate data sets to detect significant mean differences. Its statistical power has been reported to be equivalent to Tukey's post hoc test (*Ryan, 1960*), and it can be used regardless of the data distribution (*Ryan, 1960*). Statistical significance was accepted at $p < 0.05$.

## RESULTS

Figure 2A shows the CoT-*s* relationships obtained under the two conditions. In walking, significantly lower CoT values were observed with BWS than with NW at speeds faster than 1.56 m·s$^{-1}$ ($F = 16.019$, $p = 0.002$, $\eta^2 = 0.178$; Fig. 2A). In running, the CoT values were significantly lower with BWS than with NW at all speeds ($p < 0.001$). The mean EOTS was significantly faster with BWS than with NW ($2.304 \pm 0.141$ vs. $2.031 \pm 0.086$ m·s$^{-1}$; $F = 32.326$, $p < 0.001$, $\eta^2 = 0.327$; Fig. 2B). The same was the case with the PTS ($2.068 \pm 0.153$ vs. $1.932 \pm 0.075$ m·s$^{-1}$; $F = 43.209$, $p < 0.001$, $\eta^2 = 0.331$; Fig. 2B). A post hoc test revealed that the PTS was significantly slower than the EOTS under both conditions ($F = 7.865$, $p = 0.010$; Fig. 2B).

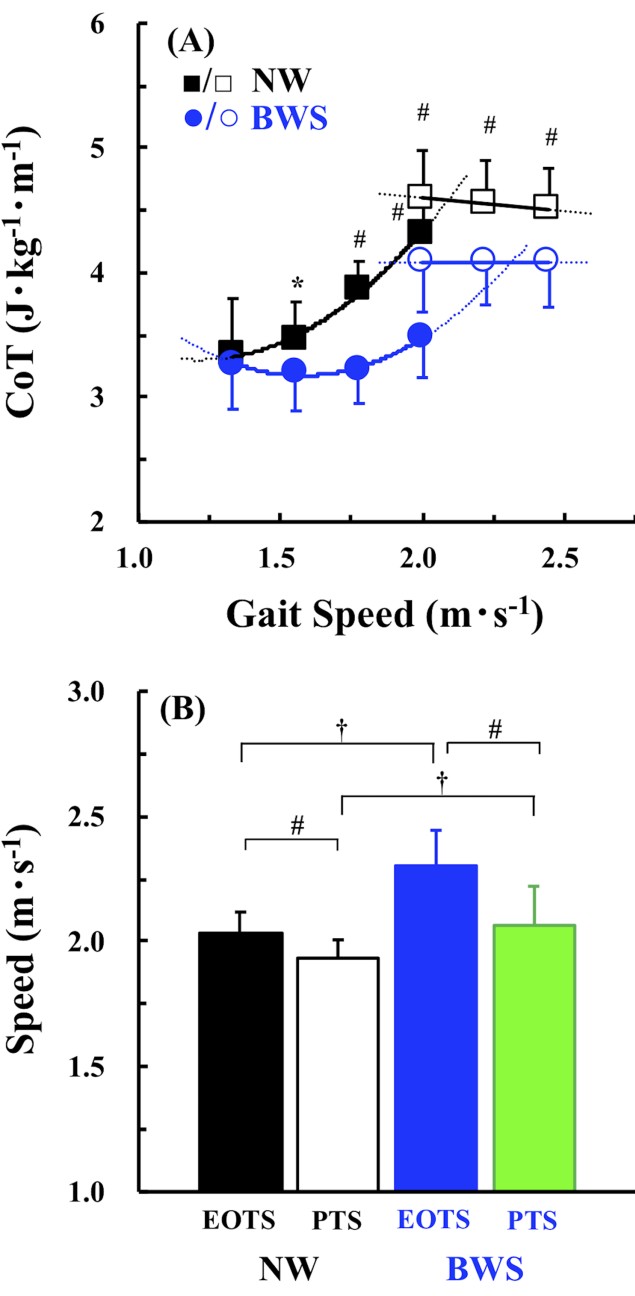

**Figure 2 Comparisons of CoT values under normal walking (NW) and body weight support (BWS) conditions.** (A) Cost of transport (CoT) as a function of gait speed under NW (black solid and open squares) and BWS (blue solid and open circles). Solid lines are the approximations using actually observed CoT values. Dotted lines are the extrapolations. (B) Comparisons of energetically optimal transition speed (EOTS) and preferred transition speed (PTS) under NW (black–white bars) and BWS (blue–green bars). Values are mean ± SD. $^*p < 0.05$, $^\#p < 0.01$, and $^\dagger p < 0.001$.

Comparisons of activities of the dorsiflexor (TA) and the combined plantar flexor activities (the average for MG, LG, and SOL) are shown in the upper panels of Fig. 3. TA activity significantly decreased after the walk–run transition under both conditions ($F = 12.896$, $p = 0.004$, $\eta^2 = 0.183$; Fig. 3A). Muscle activities of the combined plantar

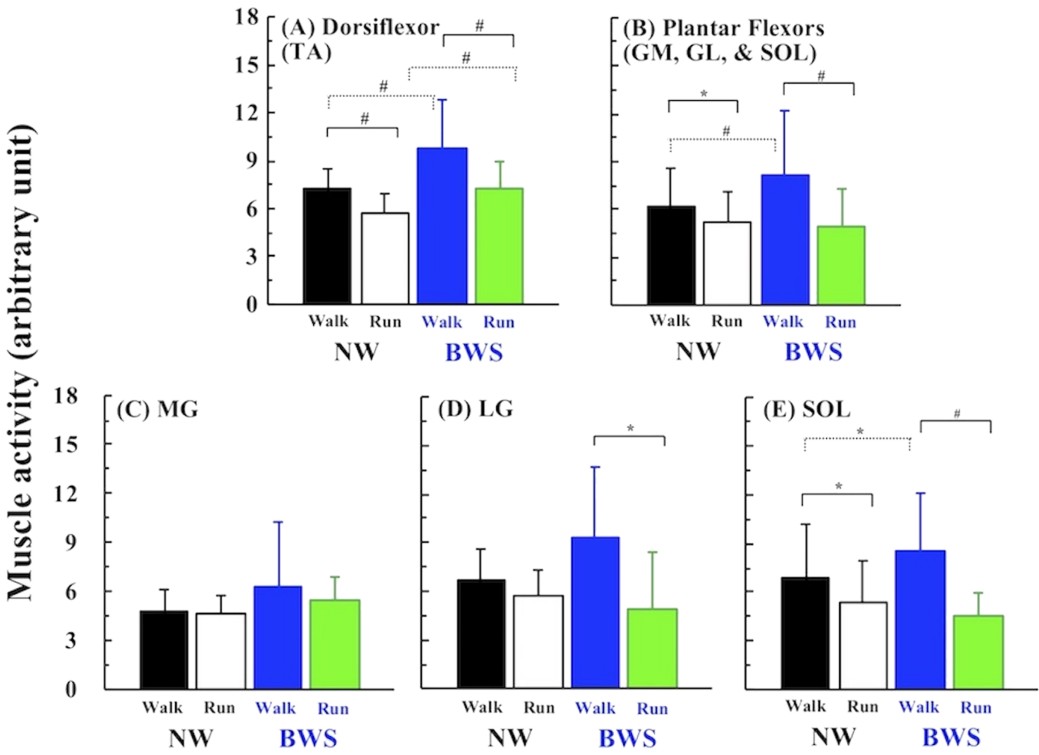

**Figure 3 Comparisons of muscle activities.** Comparisons of dorsiflexor (A; *tibialis anterior*, TA) and average of a complex of the entire plantar flexors (B). Lower panels presented each muscle activity of the *gastrocnemius medialis* (C; MG), *gastrocnemius lateralis* (D; LG), and *soleus* (E; SOL) before and after the walk-run transition under the NW and BWS. Values are mean ± SD. $^*p < 0.05$ and $^{\#}p < 0.01$.

flexors also decreased after the walk-run transition with both BWS ($F = 57.091$, $p < 0.001$, $\eta^2 = 0.111$; Fig. 3B) and NW ($F = 5.177$, $p = 0.026$, $\eta^2 = 0.019$; Fig. 3B). The lower panels of Fig. 3 show each muscle activity of the synergic plantar flexors; only SOL activity became significantly lower after the walk-run transition under both conditions ($F = 33.539$, $p < 0.001$, $\eta^2 = 0.203$; Fig. 3E).

Figure 4 shows comparisons of the MPF values for each muscle. The MPF of the TA was significantly lower after the walk-run transition under both conditions ($F = 7.484$, $p = 0.019$, $\eta^2 = 0.028$; Fig. 4A), whereas the MPF of the MG was significantly higher after the walk-run transition only with the BWS ($F = 10.516$, $p = 0.004$, $\eta^2 = 0.067$; Fig. 4B). No significant differences were observed for the LG ($F = 0.125$, $p = 0.730$, Fig. 4C). The SOL showed significantly lower MPF values after the walk-run transition under both conditions ($F = 22.355$, $p < 0.001$, $\eta^2 = 0.057$; Fig. 4D).

## DISCUSSION

One of the major findings of this study was that the walking CoT values were significantly lower with BWS than with NW at speeds faster than 1.56 m·s$^{-1}$ (Fig. 2A). The running CoT values were also lower with BWS than with NW at all speed measured (Fig. 2A); this was consistent with the results of previous studies (*Raffalt, Hovgaard-Hansen & Jensen, 2013*; *Teunissen, Grabowski & Kram, 2007*; *Pavei, Biancardi & Minetti, 2015*;

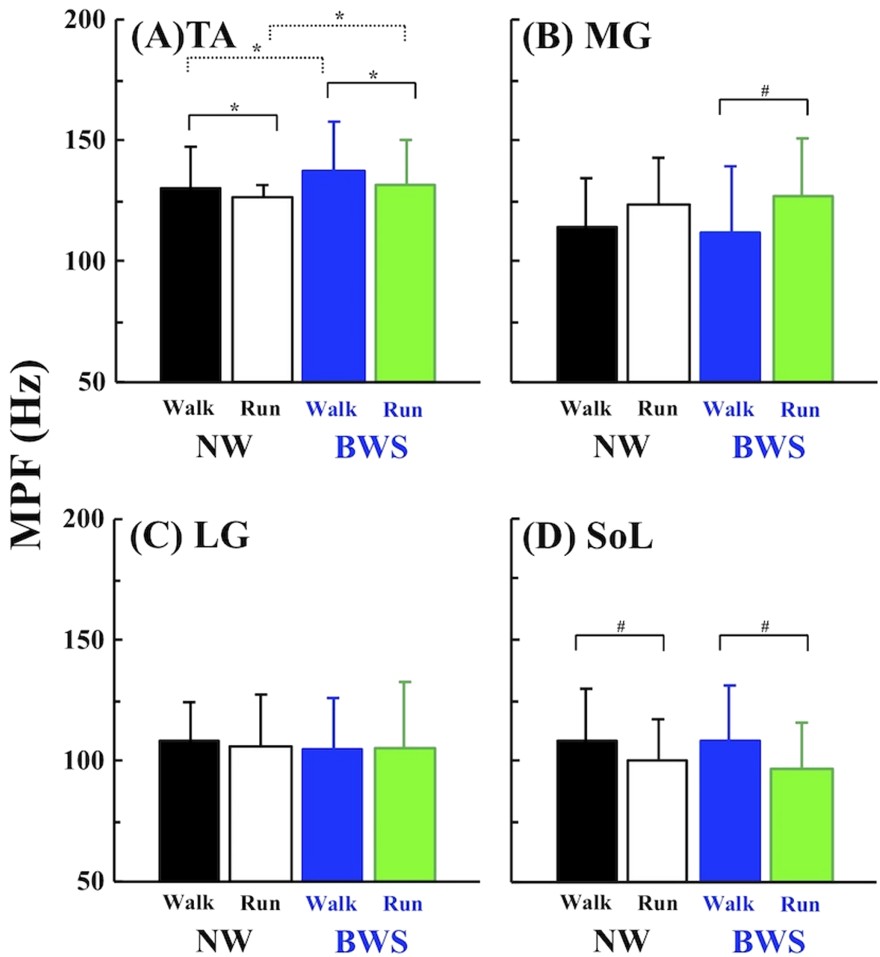

**Figure 4 Comparisons of mean power frequency (MPF).** Comparisons of MPF in the lower leg extremities before and after gait transition under NW and BWS. (A) TA: *tibialis anterior*; (B) GM: *gastrocnemius medialis*; (C) GL: *gastrocnemius lateralis*; and (D) SOL: *soleus*, respectively. Values are mean ± SD. *$p < 0.05$ and #$p < 0.01$.               

*Pavei & Minetti, 2016*; *Grabowski, Farley & Kram, 2005*; *Grabowski & Kram, 2008*). These results also demonstrated that the EOTS was faster in BWS than in NW (Fig. 2B). Although it has been proposed that the walk-run transition minimizes whole-body EE (*Cavagna & Kaneko, 1977*; *Minetti, Ardigò & Saibene, 1994*), other previous studies have found that the PTS was 5–6% slower than the EOTS in humans (*Hreljac, 1993*; *Rotstein et al., 2005*; *Tseh et al., 2002*), suggesting that the other factors may be related to the walk-run transition. The results of the present study were consistent with those of the latter studies (*Hreljac, 1993*; *Rotstein et al., 2005*; *Tseh et al., 2002*), with the PTS being significantly slower than the EOTS by 5.2% under the NW condition and 11.4% under the BWS condition (Fig. 2B). These results indicate that the walk-run transition is not triggered exclusively to minimize whole-body EE.

In support of our first hypothesis, TA activity and the associated MPF decreased when the gait pattern was switched from walking to running under both conditions (Figs. 3A and 4A). These results were equivalent to those of our recent studies that

measured TA activity during walking and running at the EOTS (*Abe, Fukuoka & Horiuchi, 2017*; *Abe et al., 2018*), which suggested that the motor unit recruitment pattern of the TA shifted more toward Type I (slow twitch) fibers rather than Type II (fast twitch) fibers after the walk-run transition. Several studies have also reported that an abrupt increase in TA activity associated with the walk-run transition (*Hreljac, 1995*; *Hreljac et al., 2008*; *Bartlett & Kram, 2008*; *Malcolm et al., 2009*; *Shih et al., 2016*). However, a careful consideration is necessary, because the TA does not contribute to producing mechanical power for forward acceleration. It is the ankle plantar flexors (MG, LG, and SOL), that play an essential role in forward acceleration during walking (*Francis et al., 2013*; *Franz & Thelen, 2016*; *Gottschall & Kram, 2003*) and running (*Hamner, Seth & Delp, 2010*). Indeed, the MG might be a potential trigger for the walk-run transition, because its power production has been shown to decrease when walking faster (*Farris & Sawicki, 2012*; *Neptune & Sasaki, 2005*). However, these previous results were based on simulated calculations; functional differences in other synergic ankle plantar flexors remain to be elucidated during human locomotion. We found that the combined activities of the plantar flexors (the average of MG, LG, and SOL activities) were significantly lower during running than walking under both conditions (Figs. 3A and 3B). These results clearly suggest that the plantar flexors also contribute to triggering the walk-run transition.

Considering muscle activity between before and after the walk-run transition, LG activity decreased significantly after the walk-run transition only under the BWS condition (Fig. 3C), and the MPF of both MG and LG was not significantly different between before and after the walk-run transition, except for the MPF of the MG under the BWS (Figs. 4B and 4C). These results were partly consistent with the results of our recent studies (*Abe, Fukuoka & Horiuchi, 2017*; *Abe et al., 2018*). The inconsistency in MG and LG activities can be explained by SOL activity. The functional role of the SOL has not been as well investigated as that of the MG and LG complex. In fact, the MGs and SOL are anatomically independent. The anatomical volume of the SOL is much larger than that of the MG and LG complex, but the SOL exerts more mechanical power in the upward direction (*Franz & Thelen, 2016*). Thus, the SOL should contribute to supporting the body during walking. Another study reported that a difference of the functional role between the MGs and SOL mainly appeared in the late stance phase (*Zajac, Neptune & Kautz, 2003*). The SOL in association with the MG and LG complex support the body in the early and middle stance phases (*Neptune, Kautz & Zajac, 2001*), but the push-off is mainly executed by the SOL only in the late stance phase (*Zajac, Neptune & Kautz, 2003*). These previous studies suggest that the SOL executes multi-tasks in one gait cycle: forward acceleration of the trunk in the late stance phase and body support in the early and middle stance phases.

One of the major limitations of our experimental approach was that we could not quantify in vivo Achilles tendon behavior. The ankle torque produced by the plantar flexors increases as walking speed increases. However, the change in length of the SOL and its tendon complex has been observed to decrease drastically during walking around the PTS (*Lai et al., 2015*), suggesting an extreme increase in the shortening length of the SOL itself in a gait cycle. This potentially increases the force production by the SOL to compensate for the decreased length change of the SOL and tendon complex in a gait

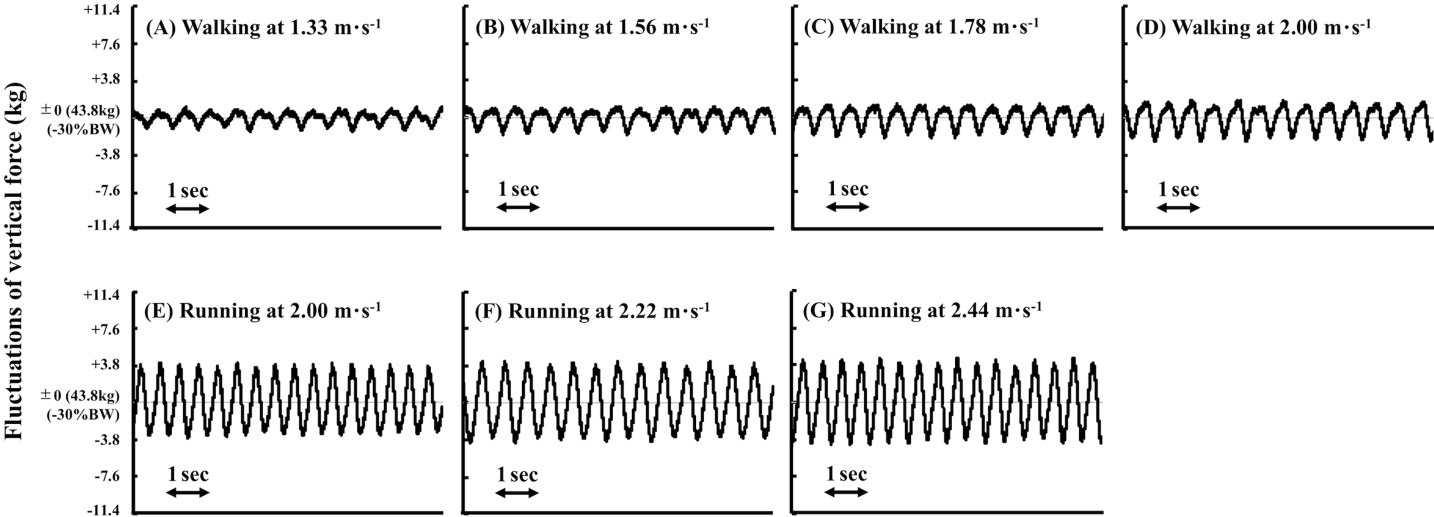

**Figure 5** **Time-vertical force relationship using body weight support (BWS) apparatus.** (A) Representative example of fluctuations of the vertical force during walking at 1.33 m·s$^{-1}$, (B) 1.56 m·s$^{-1}$, (C) 1.78 m·s$^{-1}$, and (D) 2.00 m·s$^{-1}$ in the BWS condition. During running in the BWS condition at (E) 2.00 m·s$^{-1}$, (F) 2.22 m·s$^{-1}$, and (G) 2.44 m·s$^{-1}$ are shown in the lower panels. BW means body weight.

cycle. After the walk-run transition, length change of the SOL abruptly increased (*Lai et al., 2015*), indicating that the SOL force production and related activations can be reduced. This finding was supported by our result for changes in SOL activity under both conditions (Fig. 3E). In addition, the MPF of the SOL decreased at the walk-run transition (Fig. 4D), indicating that the SOL plays a role not only in supporting the body but also in triggering the walk-run transition. Given these complex observations, the ankle plantar flexors may all contribute to the required forces, in association with changes to each motor unit recruitment pattern, to avoid the early onset of localized muscle fatigue.

Methodological considerations are necessary, because we found a faster PTS in BWS than that in NW, which contradicted numerous previous results (*Kram, Domingo & Ferris, 1997*). This conflicting result could be attributed to a difference of the used apparatus. The BWS apparatus used in some previous studies retained much larger stretch in the elastic component compared to our apparatus (*Kram, Domingo & Ferris, 1997*; *Pavei, Biancardi & Minetti, 2015*; *Pavei & Minetti, 2016*), and small vertical movements of the participants using our apparatus could result in fluctuations of the vertical force. Figure 5 shows a representative example of fluctuations of the vertical force (weight) during both gaits in the BWS condition. The baseline represents 70% of this participant's BW (30% reduction of his BW from 63.2 kg to 43.8 kg), so the suspended weight is 18.8 kg. As the fluctuation of the vertical force was about ±4 kg during running, it was fluctuated between 24% BW and 36% BW during running. During walking, the fluctuation of the vertical force was within ±2 kg, indicating that it was fluctuated between 27% BW and 33% BW during walking. These fluctuations of the vertical force are not negligible, so we used the term BWS rather than simulated reduced gravity. In the previous studies, the PTS became slower as a function of reduced body weight level (*Kram, Domingo & Ferris, 1997*; *Pavei, Biancardi & Minetti, 2015*; *Pavei & Minetti, 2016*),

which can be predicted from the Froude number (*Alexander, 1989*). The spring of our apparatus is much shorter (only 30 cm) than that of previously used apparatus due to a limitation of the room height. Thus, our spring-like apparatus yielded relatively large fluctuations of the vertical force. This intrinsic characteristic of our apparatus could reduce the mechanical power output for the vertical direction, which should be related to the decreased whole-body EE during running and walking at faster speeds (Fig. 2B), resulting in a faster PTS in BWS than in NW.

There had been a debate whether the spectral properties of the surface EMG provide information about motor unit recruitment strategies and muscle fiber type (*Von Tscharner & Nigg, 2008*; *Farina, 2008*). In recent years, many articles provided the motor unit activity using power spectral properties (MPF) of the surface EMG during dynamic exercise in vivo (*De Luca et al., 2015*; *Kallio et al., 2013*; *Von Tscharner et al., 2018*) along with the Henneman's size principle (*Henneman, Somjen & Carpenter, 1965*). To the best of our knowledge, Henneman's size principle cannot be applied during instantaneous heavy power output (*Moritani, Oddsson & Thorstensson, 1990*) and/or instantaneous eccentric contraction (*Smith et al., 1980*; *Nardone & Schieppati, 1988*; *Kallio et al., 2013*). Our experimental condition partially met such specific conditions, suggesting that alterations of the MPF values before and after the walk-run transition reflected a shift toward more Type I motor unit recruitment. That is, the walk-run transition is associated with greater slow twitch fiber recruitment of the TA and SOL, perhaps to avoid early onset of localized muscle fatigue.

## CONCLUSIONS

The PTS was significantly slower than the EOTS under both conditions, indicating that the walk-run gait transition is not only triggered by the minimization of whole-body EE. In the BWS condition, both PTS and EOTS were faster due to the potential reduction of muscular activity in both the dorsiflexor (TA) and the plantar flexors, particularly in the SOL. Reductions in the activity of these muscles in association with changes to the motor unit recruitment patterns make a considerable contribution to the walk-run transition in human locomotion.

## ACKNOWLEDGEMENTS

We specially thank to Mr. Akinobu Sakamoto, Mr. Tomokazu Iwatani, Mr. Hiromichi Ikegami, Mr. Takeshi Saito, Mr. Masaru Hashimura, and Mr. Shizuo Takatoh (Takei Scientific Instruments Co. Ltd., Niigata, Japan) for customizing both body suspension apparatus and treadmill. We also thank to Hideki Kaneko and Shouichi Matsuyama (Nihon Santeku Co. Ltd., Nagoya, Japan) for their continuous technical assistance of the EMG recording and analysis.

### Funding

This study was financially supported by a Grant-in-Aid for Scientific Research from the Japan Society for the Promotion of Science (JP19K11541 to Daijiro Abe), the Japan Society

of Physiological Anthropology Research Grant for Young Scientists (to Daijiro Abe), and a Grant-in-Aid for Kyushu Sangyo University Scientific Research and Encouragement of Scientists (No. 58 to Daijiro Abe). The funders had no role in study design, data collection and analysis, decision to publish, or preparation of the manuscript.

## Grant Disclosures

The following grant information was disclosed by the authors:
Scientific Research from the Japan Society for the Promotion of Science: JP19K11541.
Japan Society of Physiological Anthropology Research Grant for Young Scientists.
Kyushu Sangyo University Scientific Research and Encouragement of Scientists: No. 58.

## Competing Interests

The authors declare that they have no competing interests.

## Author Contributions

- Daijiro Abe conceived and designed the experiments, performed the experiments, analyzed the data, contributed reagents/materials/analysis tools, prepared figures and/or tables, authored or reviewed drafts of the paper, approved the final draft.
- Yoshiyuki Fukuoka conceived and designed the experiments, authored or reviewed drafts of the paper, approved the final draft.
- Masahiro Horiuchi conceived and designed the experiments, authored or reviewed drafts of the paper, approved the final draft.

## Human Ethics

The following information was supplied relating to ethical approvals (i.e., approving body and any reference numbers):

The ethical committee of Kyushu Sangyo University approved all procedures of this study (H28-0001-1).

## Data Availability

The raw data are available in the Supplemental Files.

## Supplemental Information

Supplemental information for this article can be found online at http://dx.doi.org/10.7717/peerj.8290#supplemental-information.

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
