# Peer review of "Why do we transition from walking to running? Energy cost and lower leg muscle activity before and after gait transition under body weight support"

_PeerJ, doi:10.7717/peerj.8290_

## Round 0.1 · original submission · Major Revisions

While the reviewers generally commented positively on the manuscript, they also provided constructive and detailed feedback that should be used to further improve the manuscript. In particular, the authors should better relate their work to existing literature, justify their hypothesis and sample size in more detail, and discuss the limitations of their study.

Reviewer 1 ·

Basic reporting

Language can be improved. I have made numerous suggestions. Key references are missing as noted. Otherwise OK

Experimental design

Method of RGS is not described with adequate detail.

Validity of the findings

Data are OK, but I can't understand why they find faster PTS in RGS. It contradicts numerous previous findings but strong researchers.

Additional comments

1. Overall, this manuscript reports new evidence that whole-body metabolic energy is not the sole trigger for the walk-run transition in humans. Rather they find that local factors (muscle activity) play at least some role. That is not a particularly new finding for normal gravity, but PeerJ specifically does not require novelty for publication. Rightly, PeerJ recognizes that replication is an important part of science. With regard to normal gravity, this study mostly replicates the previous findings of other groups: Hreljac, Prilutski, Neptune etc. The authors should do a more thorough job of citing the studies that are being replicated.

2. The simulated reduced gravity (SRG) findings are potentially novel but the critical finding that the Preferred Transition Speed is faster than in normal gravity (NG) is the direct opposite of what has been found by the Cavagna lab group, the Minetti group, the Ivanenko group and the Kram group. The authors do not recognize that contradiction, reference those papers or explain why they have found the opposite trend. In contrast, the previously reported reduction in the PTS in SRG does have a theoretical basis (Froude number).

3. All of the above studies have used at least slightly different methods of body weight reduction, but the finding of slower walk-run transition speeds is consistent in those studies. The present study uses yet another apparatus. The present authors do not provide key information about their SRG apparatus that might allow a reader/reviewer to compare the new device to previous devices. I would like to know the mass of the rigid frame + sliding part. Further, it would be helpful to have a graph of the applied force vs. time. Further, the device as depicted probably allowed the person to “hang back” allowing the harness to pull forward as well as up. The spring is quite short (only 30cm) and thus I suspect that the 3-4 cm of vertical movement by the person would cause a large amount of force fluctuation. The spring constant of 5.7 kg/cm suggests that a 3 cm movement would change the force by almost 20kg!

4. On line 140, it states that the acceleration of the treadmill speed was 1.67 m/sec^2. That is amazingly fast and I think in error. If the initial speed was 1.11 m/sec, the treadmill speed would be > 2 m/sec in less than 1 second!

5. I don’t see any problems with the EMG or metabolic energy measurement methods. However, I am not an at interpreting MPF. The statistics seem OK.

6. Line 67 Why would I expect this change in EOTS since the met cost of both walking and running are reduced in RGS? In general, the logic from lines 67-70 is confusing

7. L141 how was the gait detected/quantified?

Minor Comments/Suggestions
Line 1 Title: Why do we transition (not transit)
L17 energetic cost of transport
L18 literature (no s)
L19 argues
L20 cut: “the”
L21 body weight (not body mass)
L22 muscle activity (you do not measure work)
L24 I suggest you are consistent and use “s” for speed rather than using both s and v
L26 during constant acceleration protocol (i.e. velocity ramp)
L26 1.67 can’t be correct
L28 using electromyography of the primary ankle dorsiflexor
L29 after the walk-run transition
L31 both were faster under SRG than in NG.
L31 In both gravity levels, MPF decreased after the walk-run transition in the dorsiflexors and the combined plantar flexor activities, especially the soleus.

L34 The walk-run transition is not triggered solely by the minimization of whole-body energy expenditure.
L35 Gait transition is associated with reduced TA and soleus activities with evidence of greater slow twitch fiber recruitment perhaps to avoid….
L38 Human terrestrial locomotion
L39 it would be nice to cite some of the original studies (Margaria) rather than these citatiosn which are reviews of review papers.
L44 energetic
L44 s instead of v
L45 approximately linear
L50 Ziv is for racewalking, a very different gait
L51 explain the trigger for the walk-run transition
L53 should cite the Hreljac classic paper on this
L55 becoming
L58 I suggest using MG and LG. GM is often used for Gluteus maximus.
L64 muscular activity but also the required…
L67 muscular activity, it
L68 would be faster in SRG
L70 would be faster in SRG compared to NG.

L70 flexor (no s)
L70 decrease after the transition from walking …
L73 investigate the muscular activity and the motor unit…
L74 flexors
L74 under NG and SRG conditions.

L77 To determine…
L79 measurements
L92 & L98 body weight (BW)
L98 spring system.
L98 The set-up allowed normal leg swing.
L103 on a level…
L104 After measuring the mLetabolic cost of walking…
L105 Cut “running”
L105 with 1 min standing rests between the running stages.
L106 At all stages…
L107 and the same model of shoes in the appropriate sizes.

L109 dots over VO2 and VCO2
L111 Calibrated gas concentrations
L114 and 115 “s” instead of v
L134 to allow placement of the pre-amplified…
L137 The electrode leads were secured…
L140 increased at an acceleration of xxx
L142 throughout
L145-6 artifact in the EMG
L151 A foot sensor
L155 Subsequently, a fast Fourier…
L159 normalized to

L177 at all speeds
L178 significantly faster
L181 significantly slower
L182 the combined plantar flexor
L185 combined plantar flexor
L189 under both gravity conditions
L192 MG
L194 LG
L203 The results also demonstrated that the EOTS was faster in SRG than in NG
L210 significantly slower
L211 These results indicate that the walk-run transition is not triggered exclusively to minimize whole-body EE.
L213 I would move this sentence later in the paragraph. State your own results and then put in context.
L220 fibers after the gait transition.
L222 cut: “which are the antagonists of the TA”
L225 for the walk-run transition
L229 the combined activities
L231 cut “should”
L231 contribute to triggering the gait transition. Cut “in some way”
L240 In fact, the gastrocnemius…
L242 cut “muscular activity of the”
L244 of our experimental approach was that we could not quantify in vivo…

L248, L250 what is “shortening length”? and how does it “recover”?
L255 required forces, in association…

L260 was significantly slower
L260 …indicating that the walk-run gait transition is not only triggered by the minimization of whole-body EE. In the SRG…
L262 and EOTS were faster

Reviewer 2 ·

Basic reporting

This study is well presented in terms of article structure, literature references and background provided. However, I think that authors would be to attend a some aspect for future submission:
1) Caption of figure 3 is poor and it not include a sufficient description of the illustration.
2) Table with physical characteristics of participants should be included in Participants subsection.

Experimental design

The experimental design is well defined in accordance with the research questions to be addressed by the authors. It was performed with high technical and ethical standards.
It is clear that much work has taken place, but some technical points are not clear in the current version of the manuscript and the authors would to attend the comments included in the document attached before submitting the new version of the manuscript.

Validity of the findings

The results and discussion sections are well presented and in accordance with the aims of the manuscript. However, a detailed description of the limitations of this study and future work should be included by the authors into the discussion.

Additional comments

This manuscript presents the evaluation of the energy cost of transport and EMG during walking and running conditions to study the mechanisms that occur in the transition from walking to running. However, I think the authors need to address a few queries included in the document attached to improve the quality of the manuscript.

Annotated reviews are not available for download in order to protect the identity of reviewers who chose to remain anonymous.

·

Basic reporting

Language is generally clear and proper/professional English although there are numerous small typos and some cases of poor grammar, e.g. Lines 54-55: "recent studies showed a decrease in muscle activity of the TA in association with its motor unit recruitment pattern became lower", and Line 101 I could not determine what the abbreviation ES referred to.

Literature cited was appropriate in terms of breadth and depth although I think additional citations are needed to support the hypotheses and some of the discussion and interpretation on the EMG (please see more specific comments below on the hypotheses and in Box 3 below on EMG).

Structure was appropriate.

Appropriate testable hypotheses were posed although I was unclear how they related to the research question emphasized in the Introduction of further clarifying the triggers of the gait transition. The Introduction would benefit from greater clarity in explaining the expected results. Formal scientific hypothesis has two parts: an expected result and an explanation why that result is expected. The former part was clear to me but the latter was not.

For example, the Introduction focused on how some studies suggest TA is the muscle-level trigger for gait transition while others suggest it is the plantarflexors. How is an experiment with manipulated body weight support expected to clarify this issue, e.g. is it expected that body weight support would preferentially decrease the demand on the plantarflexors, which are major contributors to body weight support in both walking and running (Neptune et al., 2004; Hamner et al., 2010), vs. the TA which is in most studies not a major contributor to support?

Relatedly, little was made of the finding that the PTS and EOTS seemed to be much closer to each other without the simulated reduced gravity. The discussion commented on this briefly (Line 210) but this comparison was not tested statistically (I think it should be) and was not discussed extensively in relation to the muscle-level results. I think this is an important point to discuss towards the goal of clarifying why humans change gaits. It would also help to raise this point first in the Introduction to motivate testable hypotheses, so that the discussion then circles back to it later.

Experimental design

In the sample size justification, please justify the effect size, i.e. why is 0.30 sufficient for this research question, beyond a citation to Cohen. It was also not clear what statistical test this effect was for. It may be helpful to report the minimum detectable effect of this sample size in terms of the actual outcome variables (e.g. what was the minimum detectable difference in speeds, or metabolic costs?) as readers in the field will have a better sense for whether these differences are "small", "large" etc.

In most of the cited gait transition studies, comparisons were made of measurements taken at the subject's actual transition speeds, i.e. walking vs. running at 100% of the walk-run transition speed, and data are measured for walking and/or running at an overlapping range of speeds around the transition speed to determine variables like the EOTS. Here the protocol seemed considerably different: in the first part, subjects walked at several speeds and ran at several speeds, only one of these speeds was performed at both gaits (2.0 m/s), and the EOTS appeared to have been determined by extrapolative curve-fitting since it was faster at least on average than the fastest-tested walking speed in either condition. This was concerning because there is often considerably scatter in experimental data around the typically-assumed quadratic relationship between walking speed and metabolic rate. This point is difficult to address without a large effort of collecting new data but at a minimum it should be discussed in terms of implications/limitations/sensitivity.

For example, concerning the quadratic relationship and fitting of the experimental data, some recent studies have suggested fairly convincingly that the metabolic rate vs. speed relationship for running is not linear and as a result the metabolic cost of running is not invariant of running speed at least in some runners (Steudel-Numbers & Wall-Scheffler, 2009; Batliner et al., 2017). Did the authors consider also fitting a nonlinear to model to the running data and does this affect results?

In the second part of the protocol, a ramped increase in treadmill speed of 1.67 (m/s)/s was described. Here again it was not clear if data at the actual transition speed were obtained or if the speed was steady for an appreciable period of time. The text read as if the treadmill steadily accelerated at 1.67 m/s^2. I think readers will need to see more details on this protocol to be clear on what was done and what data were obtained.

Validity of the findings

The interpretation of the changes in EMG data seemed overstated. While this was not stated directly, it appeared to me that these results were interpreted as evidence of change in the type of motor units recruited, assuming the classic Size Principle holds. If this is assumed in interpreting the data, this assumption should be stated clearly in the text with supporting references. To my understanding, the strong evidence for the Size Principle is largely from isometric contractions and it is unclear if it holds in general for dynamic contractions like those in gait (for example see review by Hodson-Tole & Wakeling, 2009).

Similarly, concerning the EMG MPF results, it is unclear and contentious if "spectral properties" of surface EMG signals indicate motor unit characteristics and recruitment patterns (see Point-Counterpoint exchange in Journal of Applied Physiology 2008 volume between Farina and Wakeling). Here as well, it is suggested the authors state this assumption directly with references to support it, and discuss its implications/limitations including contrary evidence.

---

## Round 0.2 · Major Revisions

While the revision has improved the manuscript, there are some outstanding comments that need to be addressed before the manuscript can be accepted for publication. In particular, the authors should ensure that they change the manuscript in line with their responses to the reviewers so that the additional information is also available to the reader.

Reviewer 1 ·

Basic reporting

OK

Experimental design

OK

Validity of the findings

OK

Additional comments

The authors have been very responsive to my comments.
However, their added explanation of their device now prompts me to ask that the authors re-title their paper and re-name their device. I request that throughout the manuscript, they replace “simulated reduced gravity” with “body weight support, BWS”. This is not too difficult to do and I feel it is more accurate description of the device. It is a simple way to avoid appearing in contrast to previous studies of W-R transition in devices that more closely simulate RG by minimizing fluctuations in the upward applied force. The authors may also wish to change Normal Gravity (NG) to simply Normal Walking, NW. The authors should in any case use the phrase “in both conditions” rather than “in both gravity levels”. I suggest that they use the search function of their word processor and look for each instance of “gravity” and correct accordingly. The authors should also search for RGS, NG in both the text and their Figure captions and Figures.

Specific suggestions
L17 …(CoT) has been suggested as an explanation for the walk-run transition in human locomotion.
L19 triggers
L26 I still think the units for acceleration are not correct. If initial velocity was zero and a = 0.278 m/s^2, then in 60 seconds (1 minute), the velocity would be (v = at), 60 x 0.278 or 16.68 m/sec (faster than Usain Bolt). If 1km/hr per minute is correct, then the authors should state that acceleration was 0.278 m/sec/minute or calculate m/s^2 correctly.
L40 …in facilitation of economical energy expenditure.
L44 In walking, there is …
L46 … is approximately constant
L49 5-6% faster
L60 These findings suggest that
L78 plantar flexors (plural)
L126 The relationship between CoT values…
L134 only the faster one…
L143 …tape to avoid disturbing the locomotor tasks.
L146 …to visually detect when the participant…
L148 cut sentence: Note that the values…
L152 fix 0.278 again
L161 Please specify the gain. “The EMG signals were amplified with a gain of XXX (BA 1104B, Digitex….”
L165 …was applied to the stored EMG signals
L165 mean power frequency (spelling)
L166 … Wakeling, 2004). An alteration of the MPF…
L186 COT-s (not v)
L186 under the two conditions.
L219 5-6% slower
L238 Indeed, the MG
L246 Considering muscle activity…
L284 add a sentence …the vertical force. Thus, we use the term body weight support (BWS) rather than simulated reduced gravity. In these previous studies…
L288 …yielded relatively large fluctuations of the vertical force. This intrinsic characteristic of our apparatus could reduce the mechanical power output…

Reviewer 2 ·

Basic reporting

Authors have provided a detailed response for my suggestions.

Experimental design

No comments.

Validity of the findings

No comments.

Additional comments

No comments.

·

Basic reporting

no comment

Experimental design

no comment

Validity of the findings

no comment

Additional comments

I think there are still some major issues with the manuscript that need to be addressed through revision to the content of the paper, not just through responses to the reviewer:

- The sample size is not justified. It should be justified using an effect size that is meaningful for this particular area of study, not an arbitrary effect size that Cohen says is large. If there is no area-specific basis for the sample size, I think the authors should simply state that the study was exploratory in design and report the minimum detectable difference.

- I still have a major concern on the determination of the transition speed's metabolic rates by extrapolation. The authors misunderstood my statement in the original review on scatter in the data (I assume the authors don't actually think there is no scatter in their experimental data). The issue here is extrapolation outside the range of the measured data for either gait may inaccurately estimate an unmeasured metabolic rate, particularly at an unusual speed. No substantive change or addition to the content of the manuscript was made in response to this concern. Combined with the differences in equipment highlighted by Reviewer #1, it is difficult to be confidence the results (vs. other similar studies) are due to the independent variables vs. due to the protocol and equipment. Confidence here, at least on the element of the extrapolation, could be gained if it is shown that the results are insensitive to the method of extrapolation. The authors stated in their response that this is so but there is no evidence in the paper that this is so.

- Relatedly, the new information added on the fluctuation of the vertical force is a concern. I was unsure if I understood the function of the device exactly, but if the amount of force applied depended on the phase of the gait cycle as stated and with the stated spring stiffness, it seems that there would be fairly large fluctuations in the force during the gait cycle, up to perhaps ~ 500 N or so in running. Is this not a large departure from the paradigm of simulated reduced gravity, where the force would be expected to be constant? A graph of the force-time was requested but I did not see where or if this was included in the revision.

---

## Round 0.3 · Minor Revisions

The authors have not adequately addressed the reviewer comment regarding extrapolation of data. In their response to the reviewers, the authors indicate that extrapolation is commonly used for calculating the EOTS. However, this response is not reflected in the revised manuscript. In fact, extrapolation is not at all mentioned in the manuscript.

I would like to ask the authors to explicitly address this point in the revised manuscript, provide justification of the extrapolation of CoT data and acknowledge that EOTS is based on the extrapolation of CoT data. Also, the solid and dotted lines in Fig. 2A should be described in the figure legend.

---

## Round 0.4 · accepted · Accept

The authors have adequately addressed the outstanding comments.